# Effects of Different Stand Densities on the Composition and Diversity of Soil Microbiota in a *Cunninghamia lanceolata* Plantation

**DOI:** 10.3390/plants14010098

**Published:** 2025-01-01

**Authors:** Zuyuan Xu, Fei Fan, Qinmin Lin, Shengzhou Guo, Shumao Li, Yunpeng Zhang, Zhiyi Feng, Xingxing Wang, Christopher Rensing, Guangqiu Cao, Linkun Wu, Shijiang Cao

**Affiliations:** 1College of Forestry, Fujian Agriculture and Forestry University, Fuzhou 350002, China; 18143582739@163.com (Z.X.); gsz19559162600@126.com (S.G.); 13355632751@163.com (X.W.); cncgq@126.com (G.C.); 2College of Jun Cao Science and Ecology (College of Carbon Neutrality), Fujian Agriculture and Forestry University, Fuzhou 350002, China; 18960538816@163.com (F.F.); 13063163276@163.com (S.L.); zyp13519662679@163.com (Y.Z.); 3College of Life Sciences, Fujian Agriculture and Forestry University, Fuzhou 350002, China; 18760030926@163.com; 4Institute for Environmental Microbiology, College of Resources and Environment, Fujian Agriculture and Forestry University, Fuzhou 350002, China; savannah11210802@163.com (Z.F.); crensing94@gmail.com (C.R.); 5Marine Electromechanical College, Xiamen Ocean Vocational College, Xiamen 361102, China

**Keywords:** Chinese fir, plantation forest, high-throughput sequencing, soil microorganisms, microbial diversity

## Abstract

As sustainable forest management gains increasing attention, comprehending the impact of stand density on soil properties and microbial communities is crucial for optimizing forest ecosystem functions. This study employed high-throughput sequencing in conjunction with soil physicochemical analysis to assess the effects of stand density on soil physicochemical properties and microbial community characteristics in Chinese fir plantations, aiming to elucidate the influence of density regulation on ecosystem services. Our results suggested that changes in soil physicochemical properties and microenvironmental conditions were key drivers of soil microbial diversity. Total carbon (TC), soluble nitrogen (SN), and light fraction organic matter decreased with increasing stand density, while total potassium (TK) and available phosphorus (AP) concentrations increased. The plot with a density of 900 trees ha^−1^ exhibited the highest bacterial diversity, in contrast to the plot with 1500 trees ha^−1^, which showed the lowest. The dominant microbial taxa were similar across different stand retention densities, with *Acidobacteria*, *Proteobacteria*, and *Chloroflexi* being the predominant bacterial phyla and *Ascomycota* and *Basidiomycota* being the main fungal groups. Significant positive correlations were observed between soil microbial community structures and environmental factors, particularly with respect to soil phosphorus and nitrogen content. The present study demonstrated that reduced stand densities modulated soil nutrient content and enhanced bacterial diversity, thereby contributing to a more complex and stable soil ecosystem structure. These insights provide a scientific foundation for optimizing the management of Chinese fir plantations, thereby supporting the sustainable development of forest ecosystems.

## 1. Introduction

Chinese fir (*Cunninghamia lanceolata*) is an economically important tree species that is widely planted in the south of China [1]. It plays a critical role not only in timber production but also in ecosystem services such as soil conservation and carbon fixation [2]. However, large-scale monoculture plantations of fir plantations and continuous woodland management patterns have led to a number of ecological problems, including increased pests and diseases, declining soil fertility, and soil acidification [3]. Stand density is a measure of how densely distributed trees are in a forest, and it describes quantitatively the competition in a forest stand (i.e., the population of trees in a given area). In the process of forest management, stand density is a vital and direct regulatory factor that directly affects the growth space of forest trees and resource utilization efficiency [4]. Forest trees have high requirements for environmental factors, such as soil nutrients and water and light during early growth, and excessive stand density may lead to increased competition for resources and affect plant growth. However, excessive low stand densities may lead to soil nutrient loss and biodiversity decline, which in turn may reduce ecosystem services [5]. A large number of studies have been reported on the effects of stand density on forest management, focusing mainly on the areas of ecological function, forestry management, climate change and carbon storage, and biodiversity conservation [6,7], focusing mainly on the areas of ecological function, forestry management, climate change and carbon storage, and biodiversity conservation [8,9,10]. In the field of forest ecology, the effect of stand density on the growth of tree species, diversity of understory vegetation, and soil properties is an important research topic. It has been found that Chinese fir at a stand density of 650 trees ha^−1^ promotes the growth of understory vegetation, increases species diversity and biomass, and at the same time improves the organic matter content and fertility retention capacity of the soil, which has a positive effect on the stability and sustainable development of willow forests [11]. For *Pinus tabulaeformis*, the understory vegetation diversity index shows a decreasing trend with the increase in stand density. The low stand density of the plantation forest of *Pinus tabulaeformis* is conducive to maintaining the stability of the community structure and increasing the diversity of understorey species [12]. The results of these studies provide a scientific basis for forest management and guide the rational adjustment of stand density to maximize ecological and economic benefits. Therefore, reasonable stand density is essential for improving the ecological environment and species diversity of plantation forests and contributing to community succession from plantation forests to natural forests. Meanwhile, studies have focused on the effects of stand density on soil microbial diversity and function [13,14,15]. Soil microorganisms are not only involved in soil nutrient cycling but also an important force in maintaining ecosystem stability. Studies have shown that stand density indirectly affects soil nutrient cycling and ecosystem services by influencing the structure and function of soil microbial communities [16]. Moderate stand density facilitates the growth of key functional bacteria, such as cellulolytic and ammonifying bacteria, which play an essential role in soil organic matter decomposition and nutrient conversion [17]. Higher-density stands may increase microbial diversity and abundance by providing more organic matter and habitat, promoting the growth of nitrogen-fixing and decomposing bacteria, and enhancing the nitrogen and carbon cycles. Nevertheless, excessive densities may lead to competition for resources and may affect microbial community structure and function [18]. Moderate stand density promotes soil microbial diversity, which in turn has long-term positive effects on ecosystem services such as soil nutrient retention, carbon storage, and biodiversity conservation. It has been found that moderate stand density favors the growth of key functional flora, such as cellulolytic and ammonifying bacteria, which play an important role in soil organic matter decomposition and nutrient conversion [19]. Soil microbial communities are critical for maintaining energy flow, material cycling, and information exchange in forest ecosystems. They are central drivers of the soil carbon cycle, which affects plant productivity and soil ecosystem stability [20,21]. Soil microbial communities are extremely sensitive to environmental changes, and their composition and structure are influenced by factors such as habitat, soil physicochemical properties, and human activities [22]. At the same time, vegetation types and geographical differences have received widespread attention for soil microbial diversity and stability [23,24,25].

In forest ecosystem management, stand density is recognized as a crucial factor influencing soil microbial diversity and function in fir plantations. However, although studies have been conducted to explore soil microbial communities in fir plantation forests, relatively few studies have addressed soil microbial diversity and function under different stand densities. In particular, how soil microbes respond to and adapt to the environment under different density conditions has rarely been reported [26]. As a result, further study is needed to be carried out in this area. In this study, five fir woodlands with different stand densities were selected to analyze the effects of changes in stand density on soil microbial diversity using high-throughput sequencing and how these changes further affect the structure and potential functions of soil microbial communities. Through this study, we expect to optimize forest management strategies, improve soil quality and productivity, and enhance the sustainable management of fir plantation forests. This will contribute to a deeper understanding of the roles of soil microbes in maintaining nutrient cycling and ecosystem stability.

## 2. Results

### 2.1. Differences in Soil Physicochemical Properties of Fir Plantation Forests with Different Stand Densities

Soil physicochemical properties within the 0–20 cm depth were analyzed across various stand density treatments (Table 1). While soil pH, total nitrogen (TN), and readily available potassium (AK) varied among stand densities, these differences were not statistically significant. Soil pH remains consistently acidic across all plots, ranging from 4.36 to 4.73. Soil organic carbon (SOC) and available phosphorus (AP) were significantly higher at a stand density of 2505 trees ha^−1^ when compared to other densities. In contrast, total carbon (TC), total nitrogen (TN), soluble nitrogen (SN), and available potassium (AK) were significantly higher at a stand density of 900 trees ha^−1^. Total potassium (TK) content was observed to be highest at stand densities of 1500 trees ha^−1^ and 2100 trees ha^−1^. Total phosphorus (TP) was highest at 1500 trees ha^−1^, with no significant differences observed among the remaining densities.

### 2.2. Soil Microbial Community Structure in Fir Plantation Forests with Different Stand Densities

#### 2.2.1. OTU Cluster Analysis

The Venn diagram clearly reflects the number of shared and unique OTUs among the samples (Figure 1). As shown, clustering analysis of sequences obtained by high-throughput sequencing revealed a total of 1347 shared bacterial OTUs and 526 shared fungal OTUs across the five stand retention density treatments. As for the bacterial community, the number of OTUs specific to a density of 900 trees ha^−1^ was the highest, totaling 613. The next highest was at a density of 2100 trees ha^−1^, which had 243 specific OTUs, while the lowest was observed at 1500 trees ha^−1^, with only 105 specific OTUs. The stand density of 900 trees ha^−1^ supported more unique bacterial species, while the number of endemic OTUs decreased as stand density increased, which may imply that higher-density stands may have a negative effect on the diversity of certain bacterial species. As for the fungal community, the number of OTUs specific to 1200 trees ha^−1^ was the highest, totaling 465, while the next highest was at 900 trees ha^−1^, with 264 specific OTUs. The number of endemic OTUs decreased with increasing stand density, especially under the 1500 trees ha^−1^ treatment, which had the lowest number of endemic OTUs at 144. The number of OTUs for both soil bacteria and fungi was the lowest under the 1500 trees ha^−1^ treatment, suggesting that excessive stand density may inhibit soil microbial diversity. This indicates that different stand densities may affect the structure of the soil microbial community, leading to differences in the number of OTUs specific to each density.

#### 2.2.2. Differences in the Phylum-Level Structure of Soil Microorganisms

Although the dominant flora were similar across the different stand density samples, there were significant differences in the abundance of dominant bacteria (Figure 2). This suggests that even in ecosystems with similar species composition, environmental conditions such as stand density significantly affect microbial abundance. Based on the results of species annotation, the ten most abundant bacterial phyla were *Acidobacteriota*, unidentified_*Bacteria, Proteobacteria*, *Chloroflexi*, *Actinobacteria*, *Firmicutes*, *Verrucomicrobiota*, *Myxococcota*, *Actinobacteriota*, and *Bacteroidota*. *Acidobacteriota* consistently had the highest relative abundance across all stand density conditions, implying that these taxa are particularly dominant in these environments or more adaptable to environmental change. *Firmicutes* had the highest abundance at 1500 trees ha^−1^, with the lowest abundance at 1200 trees ha^−1^, and *Proteobacteria* had the highest abundance at 900 trees ha^−1^. The predominant fungal phyla ranked by relative abundance were *Blastocladiomycota*, *Chytridiomycota*, *Rozellomycota*, *Mucoromycota*, *Zoopagomycota*, *Mortiorellomycota*, GS01, *Glomeromycota*, *Basidiomycota*, and *Ascomycota*. Among these, *Ascomycota* was the dominant fungal group across all stand density conditions; however, its abundance was significantly lower at 2505 trees ha^−1^ compared to the other stand densities. The taxa of *Zoopagomycota* were predominantly observed in conditions with 2100 trees ha^−1^, while GS01 bacteria were found in conditions with 1500 trees ha^−1^.

#### 2.2.3. Differences in Genus-Level Structure of Soil Microorganisms

At the genus level, the dominant floras across different stand density samples were similar, but the abundance of dominant bacteria varied significantly (Figure 3). This suggests that environmental conditions, such as stand density, may influence microbial abundance even if species composition is similar. Specifically, among bacterial genera, the relative abundance of *Candidatus-Solibacter* was higher than that of other species, while among fungal genera, the relative abundance of unidentified *Venturiales*-sp was the highest. This suggests that these genera play important roles in ecosystems or possess a greater adaptability to varying environmental conditions. The higher abundance of strains in 1500 trees ha^−1^, along with the elevated presence of *Escherichia-Shigella* and *Lactobacillus* in these plots, may indicate that the soil conditions at this density are particularly conducive to the growth of these genera. The high relative abundance of *Gardnerella* at the 2505 trees ha^−1^ may be linked to the specific environmental conditions present here. At the fungal genus level, *Monographella* and *Hygrocybe* exhibited significantly greater abundance in the 900 trees ha^−1^ compared to other samples, but *Pseudoplectania* was not detected in this density. *Pseudoplectania* was present in the 2100 trees ha^−1^ treatment, unidentified_GS01_sp was abundant at 1500 trees ha^−1^, and *Geoglossum* was found at 1200 trees ha^−1^.

### 2.3. Analysis of Soil Microbial Diversity in Fir Plantations with Different Stand Densities

#### 2.3.1. Alpha Diversity Analysis

Alpha diversity indices for soil microorganisms were calculated separately for different stand retention densities based on sequencing results. In this study, the Chao1, Shannon, PD-whole, and Good-coverage indices were calculated for different samples. The Shannon index and Chao indices at different stand retention densities ranked as follows: 900 trees ha^−1^ > 2505 trees ha^−1^ > 2100 trees ha^−1^ > 1200 trees ha^−1^ > 1500 trees ha^−1^. This pattern indicates that the highest diversity of bacterial populations was found at the 900 trees ha^−1^ site, while the lowest bacterial diversity was observed at the 1500 trees ha^−1^ site. In the study examining the Alpha diversity of fungal communities, it was found that the Shannon index was significantly higher than other stand densities, and the community diversity was the highest in the treatment of 1200 trees ha^−1^. The Shannon index for different stand retention densities, ranked from high to low, was as follows: 1200 trees ha^−1^ > 2100 trees ha^−1^ > 2505 trees ha^−1^ > 1500 trees ha^−1^ > 900 trees ha^−1^. Similarly, the Chao1 index ranked in the same order: 1200 trees ha^−1^ > 2100 trees ha^−1^ > 2505 trees ha^−1^ > 900 trees ha^−1^ > 1500 trees ha^−1^. These results indicate that the 1200 trees ha^−1^ treatment had the highest fungal diversity. The microbial community diversity of the 1500 trees ha^−1^ sample plot was significantly higher than in the other treatments, which may imply that 1500 trees ha^−1^ favors microbial diversity. There was no significant difference in soil microbial community diversity between 1200 trees ha^−1^ and 1500 trees ha^−1^, indicating both stand densities had similar effects on microbial diversity. Bacterial diversity was lowest at the 1500 trees ha^−1^ sample site, while Shannon’s index was significantly higher at 1200 trees ha^−1^ than in the other stand densities. Fungal community diversity was also highest at 1200 trees ha^−1^, whereas fungal diversity was lowest at 900 trees ha^−1^. This divergence in trends indicates that different microbial communities may respond differently to variations in stand density.

#### 2.3.2. Principal Component Analysis and Hierarchical Cluster Analysis

Based on the principal component analysis of high-throughput sequencing, bacterial PC1 and PC2 accounted for 39.5% and 26.8% of the variation, respectively (Figure 4). These components were identified as the main factors contributing to the differences between samples, indicating that different stand densities had a greater impact on the structure of the soil bacterial community. The bacterial communities at 900 trees ha^−1^ and 1200 trees ha^−1^ were relatively similar, suggesting that their community compositions differed minimally. In contrast, the bacterial communities from soil samples at other stand densities were more dispersed, with obvious structural differences. The contribution of fungi PC1 and PC2 was 38.1% and 27.1%, respectively, and the fungal community of the soil samples was more dispersed. Hierarchical clustering analyses revealed that bacterial and fungal communities exhibited significant clustering at a stand density of 900 trees ha^−1^. In contrast, no significant clustering trend was observed for bacterial and fungal communities in the sample plots at other stand densities.

### 2.4. Correlation Analysis Between Soil Physical and Chemical Properties and Microbial Community

As for the bacterial community, *Acidobacteriota* was positively correlated with total phosphorus. Additionally, *Proteobacteria* had a non-significant relationship with effective phosphorus (Figure 5). *Firmicutes* is positively correlated with soil urease activity, and *Cyanobacteria* is positively correlated with soluble nitrogen. Microbial biological nitrogen was significantly positively correlated with effective phosphorus, while soluble nitrogen was positively correlated with soil acid phosphatase activity. Moreover, nitrate nitrogen was positively correlated with soil sulfide concentration. There was no significant relationship between polyphenol oxidase activity and soil urease activity. As for the fungal community, there was a significant positive correlation between *Basidiomycota* and total phosphorus, whereas the correlation between *Ascomycota* and total phosphorus was not significant. The correlation between *Glomeromycota* and effective phosphorus was also not significant. *Zoopagomycota*, on the other hand, showed a significant negative correlation with dissolved organic carbon.

## 3. Discussion

### 3.1. Effects of Stand Density on Soil Physicochemical Properties

Our study revealed that TC, TN, soluble nitrogen, and AP initially decreased and then increased with increasing stand density, while TK, TP, and SC exhibited an opposite trend (Table 2). This may be attributed to increased sunlight reaching the forest floor at lower stand densities, which elevates soil temperature and promotes microbial activity and organic matter decomposition [27]. We found higher TK and TP at intermediate stand densities and higher levels of effective phosphorus in fir stands at higher stand densities [28]. The increased demand for nutrients in high-density stands likely stimulates root activity and nutrient uptake, especially for phosphorus. Furthermore, increased plant mortality at higher stand densities leads to greater inputs of organic matter. Upon decomposition by soil microorganisms, this organic matter releases nutrients such as phosphorus and potassium back into the soil [29,30]. This nutrient cycling is crucial for maintaining soil fertility at higher stand densities [31]. While some studies have reported increased phosphorus and potassium uptake by plants following canopy thinning [32], these contrasting results likely reflect differences in climatic conditions, vegetation type, soil properties, microbial activity, and recovery time among study sites [33,34].

### 3.2. Effects of Stand Density on Soil Microbial Diversity and Community Composition

The research found that the dominant bacterial phyla in the soil microbial community were *Acidobacteria*, *Proteobacteria*, and *Chloroflexi*, with *Acidobacteria* being significantly more abundant than other bacterial groups. This phenomenon may be related to the acidic characteristics of the soil at the sampling site, as *Acidobacteria* typically prefer acidic environments [35]. Regarding *Proteobacteria*, Wang et al. [36] found in their study of Chinese fir that the addition of nitrogen can increase the relative abundance of both *Proteobacteria* and *Actinobacteria* in the soil. In contrast, Cai et al. [37] demonstrated in their study of larch plantations that *Proteobacteria* have the highest relative abundance in unthinned plots. These findings suggest that bacterial phyla may respond differently to variations in forest density and are related to the type of forest. In fungal communities, the relative abundance of *Ascomycota* and *Basidiomycota* is the highest, with *Ascomycota* accounting for more than 50% of the relative abundance in most plots, making it the dominant group in the entire fungal community [38]. *Ascomycetes* and *Basidiomycetes* play a key role in the decomposition process of soil organic matter; they are involved not only in the degradation of complex organic materials but also significantly impact the decomposition of lignocellulose in plant residues. A number of key species are present in these fungal taxa that play a key role in organic matter decomposition. They are highly connected in microbial networks, and the removal of these species may lead to significant changes in microbial community composition and functioning [39,40]. In this study, although the relative abundance of dominant bacteria did not differ significantly among stand densities, the relative abundance of dominant fungal phyla differed significantly. This suggests that bacterial communities have a stronger resistance to changes in stand density than fungal communities. This difference may be related to the smaller-scale ecological niches of bacterial communities in the soil and their lower degree of symbiosis with plants [41]. Additionally, bacteria have the ability to utilize a variety of substrates, which may contribute to the stability of bacterial populations. It is noteworthy that although soil microorganisms from each sample site were similarly categorized at the phylum level, distinct differences emerged at finer taxonomic levels, such as at the genus level, where significant differences were observed [42]. For instance, the distribution of bacteria and fungi at a density of 900 trees ha^−1^ is more even, with unique genera present in each plot. Notably, the *Hygrocybe*-like fungi, which are specific to the 900 trees ha^−1^ plot and typically found in better ecological environments, are particularly prominent. In contrast, the unidentified-GS01-sp class of bacteria, which was characteristic of the sample plot with a density of 1500 trees ha^−1^, may be related to the high bacterial diversity at low stand densities [43].

Soil microbial alpha diversity can reflect the evenness and richness of microbial communities. According to the analysis of diversity indices, the Shannon and Chao indices of soil microbial bacterial communities under different stand retention densities decreased in the following order: 900 trees ha^−1^ > 2505 trees ha^−1^ > 2100 trees ha^−1^ > 1200 trees ha^−1^ > 1500 trees ha^−1^, while the Simpson index shows no significant differences across various stand retention densities. This indicates that soil microbial diversity is highest and community composition is most complex at 900 trees ha^−1^, whereas at 1500 trees ha^−1^, soil microbial diversity is lowest and community structure is simplest. This difference may be related to the effect of soil microbial activity on soil fertility. Soil microorganisms play a vital role in soil fertility by breaking down organic matter and cycling nutrients [44]. Therefore, the level of soil microbial diversity directly affects soil biochemical processes and plant growth. The higher microbial diversity at a density of 900 trees ha^−1^ may have facilitated the cycling of soil nutrients and decomposition of organic matter, thereby increasing soil fertility and plant growth potential. In contrast, at a density of 1500 trees ha^−1^, lower microbial diversity may have limited these processes, which affected soil fertility and plant growth. The Shannon and Chao1 indices of fungal communities showed different trends at different stand retention densities, which may be related to the understory vegetation type and distribution pattern. In this study, the low fungal diversity under low stand retention density could be attributed to the fact that understory vegetation under low depression is able to capture more light energy and nutrients, while the shrub layer and herbaceous layer prevented apoplastic material from falling to the ground, leading to a decrease in soil fertility, which in turn affects the metabolism and function of fungi [45]. Stand density significantly affects soil microbial communities by regulating key environmental factors such as light, temperature, and understory vegetation cover. In high-density stands, the shading effect of trees reduces light reaching the ground, leading to decreased soil temperature, which may inhibit microbial metabolic activity and growth rate [46]. In contrast, low-density stands allow more sunlight penetration, increasing soil temperature and promoting microbial activity and diversity [47]. Additionally, vegetation cover in high-density stands increases soil organic matter input, providing a rich carbon source for microbes, while low-density stands may lead to rapid decomposition of soil organic matter, affecting microbial community structure [48]. Changes in vegetation cover also affect the physical protection and water retention capacity of the soil, further regulating water and nutrient availability to microorganisms [49]. Changes in these microenvironments collectively affect the soil microbial community, influencing its diversity, composition, and function, which in turn may influence ecological processes in the soil [50].

### 3.3. Relationship Between Soil Microbial Community Structure and Environmental Factors

Correlation studies have found that NN, AP, and AK have a significant impact on bacteria at the phylum level and are the main factors affecting community structure. It is noteworthy that soil physicochemical properties have no significant effect on the *Proteobacteria*, which may be related to the diversity and adaptability of *Proteobacteria* [51]. *Proteobacteria* is the largest bacterial phylum, exhibiting great internal variability and the ability to survive and reproduce under a wide range of environmental conditions, and therefore showing high adaptability to changes in soil physical and chemical properties [52]. *Actinobacteria* is positively correlated with AP and AK. This correlation may be due to the symbiotic relationship formed between the actinomycetes and the plant root system. *Actinobacteria* are able to interact with plant roots to form a mutually beneficial symbiotic system that effectively utilizes these nutrients in the soil. Likewise, *Actinobacteria* play an important role in soil ecosystems, participating in the decomposition of organic matter and nutrient cycling [53]. The *Cyanobacteria* is negatively correlated with NN and AK. It may be due to the fact that *Cyanobacteria* are mainly photosynthesizing in the soil and have a high nitrogen requirement, which may inhibit the growth of *Cyanobacteria* when the nitrogen content in the soil is too high [54].

TP, TC, AP, dissolved organic carbon (DOC), ammoniumnitrogen (AN), and soluble acid phosphatase (S.ACP) have a highly significant impact on bacteria at the phylum level and are the main factors affecting bacteria. The dominant bacteria *Acidobacteriota* is positively correlated with TP, while *Bacteroidota* and *Myxococcota* are negatively correlated with TP. Some members of *Acidobacteriota* are recognized as plant growth-promoting bacteria (PGPB), which can help plants absorb phosphorus from the soil by producing plant growth-promoting substances (such as phytases) [55]. Bacteroidota is a class of bacteria widely distributed in marine and soil environments, playing a critical role in the decomposition and cycling of organic matter. Bacteroidota may compete with other bacteria for phosphorus resources in the soil. Their activity may inhibit other bacteria that can promote phosphorus uptake, thus affecting TP levels [56]. *Myxococcota* is a class of bacteria with a social lifestyle, existing in groups and capable of forming fruiting bodies. These bacteria may survive in phosphorus-limited environments, enhancing their survival and dispersal abilities by forming fruiting bodies [57].

As for the fungal community, *Basidiomycota* shows a significant positive correlation with total phosphorus (TP), while *Ascomycota*’s correlation with total phosphorus is not significant and shows a significant negative correlation with AN. Many fungi in *Basidiomycota*, especially ectomycorrhizal fungi, form symbiotic relationships with plant roots [58]. When the total phosphorus content in the soil is high, it can promote the growth and activity of these fungi, thereby enhancing their symbiotic relationship with plants [59]. Perhaps owing to the fact that more nitrifying bacteria and other nitrogen-cycling microorganisms may be present in soils with higher nitrogen content, and these microorganisms compete with *Ascomycota* for nitrogen sources, leading to a decrease in the relative abundance of *Ascomycota* [60]. It is essential to point out that *Zoopagomycota* shows a significant negative correlation with TC, TN, DOC, and DON. *Zoopagomycota*, by preying on soil microbes, may reduce the number and activity of these microbes, thereby indirectly affecting the content of TC, TN, DOC, and DON in the soil. This is because these microbes play a key role in the decomposition of soil organic matter and participation in the nitrogen cycle. Our study found that the correlation between soil physicochemical properties and bacteria is stronger than that with fungi, which is consistent with the results of Ren et al. [61], indicating that soil physicochemical properties have an important impact on the changes in soil microbial diversity and community composition. This is also reflected in the studies of Ni et al. and Chai et al. [62,63]. These studies found that most soil microbial communities rely on the decomposition of organic matter for energy and reach their highest content in areas with high soil organic matter content [64].

## 4. Materials and Methods

### 4.1. Overview of the Pilot Area

This experimental area is located in the Banqiao Work Area of Yangkou State Forestry Farm, Fujian Province, China, with the geographic coordinates of 117°54′26″ E and 26°47′43″ N. The region has a subtropical monsoon climate with long hours of sunshine, with an average annual sunshine duration of about 1700 h and a frost-free period of about 300 days. The average annual temperature is 18.8 °C, and the average annual precipitation is about 1700 mm (Figure 6). The study area is located on the southern slope of the mountain, with an average slope of 30 degrees. The soil type is a mountainous red loam with a high organic matter content, a depth of more than 90 cm, an acidic pH, and a stand index of 18. The Chinese fir plantation forests in the region have been established for more than 20 years, and fir is a fast-growing timber species. According to the land use classification, the area is classified as a plantation timber forest. Understory-associated vegetation is dominated by *Ficus hirta*, *Dicranopteris dichotoma*, *Maesa japonica*, *Oreocnide frutescens*, *Diplazium metteniamum*, *Pteris fauriei,* and so on. The experimental forest was established in 2008 using “020” asexual seedling cuttings at a density of 3300–3600 trees ha^−1^. In the cultivation of medium-diameter or large-diameter fir, the initial planting density is usually set at 3300–3600 trees ha^−1^, and after 4–6 years, inter-planting is carried out. In March 2013, the forest underwent thinning to optimize the stand structure. The growth of fir trees in 15 sample plots of fir plantation forests was investigated (Table 3), and we calculated the mean and standard deviation and found that there was no significant difference in the growth of fir trees of different densities.

### 4.2. Sample Design and Sample Collection

In September 2022, soil sampling was carried out in Chinese fir plantations with different stand densities. Five stand densities of 900 trees ha^−1^, 1200 trees ha^−1^, 1500 trees ha^−1^, 2100 trees ha^−1^, and 2505 trees ha^−1^ were selected for sampling, and three sample plots were randomly set up for each stand density (the area of each sample plot was 20 × 20 m), and in the upper, middle, and lower three of each sample plot, soil samples were collected in the 0–20 cm soil layer using a 5 cm diameter soil auger. After collection, the samples were put into self-sealing bags and brought back to the laboratory for subsequent determination of the physicochemical properties of the soil. The same approach was taken at a depth range of 0~10 cm, and these samples were placed in an ice box to keep them cold and ensure that microbial activity was not compromised. Upon arrival at the laboratory, these samples were quickly transferred to a −80 °C refrigerator for storage, and to ensure the reliability of the results, three replicates of each sample were made for soil microbial community analysis.

### 4.3. Determination of Soil Physical and Chemical Properties

A variety of analytical techniques were used in this study to assess the chemical properties of the soil [65]. The soil pH was determined by potentiometric method, and soil organic carbon content was analyzed by potassium dichromate hydration heating method. The determination of TN was based on the Kjeldahl method, while total soil phosphorus was determined by NaOH solution–molybdenum antimony colorimetric method. Effective phosphorus was determined using the NH_4_F, HCl leaching–molybdenum antimony colorimetric method, while fast-acting potassium was determined by CH_3_COONH_4_ leaching followed by the use of the atomic absorption spectrophotometer method. The determination of trace elements was carried out by atomic absorption spectrometry (AAS). In addition, we measured DOC, DON, and AK, as well as soil enzyme activities. The specific experimental steps and conditions were referred to the method of Wang et al. [66].

### 4.4. DNA Extraction and Sequencing of Soil Microorganisms

After extracting genomic DNA from the samples using the CTAB method, we assessed the purity and concentration of the DNA by agarose gel electrophoresis. First, an appropriate amount of extracted DNA was placed in a centrifuge tube and diluted to a concentration of 1 ng/μL using sterile water. Next, specific regions were selected for sequencing, and specific primers with Barcode were used. For PCR, we used Phusion^®^ High-Fidelity PCR Master Mix (New England Biolabs (NEB), Ipswich, MA, USA)with GC Buffer from New England Biolabs, as well as high-efficiency, high-fidelity enzymes to optimize the PCR reaction. The specific primers and amplification regions used in this study were as follows: for the bacterial 16S rRNA V3-V4 highly variable region, primers 515F (5′-GTGCCAGCMGCCGCGGTAA-3′) and 806R (5′-GGACTACHVGGGTWTCTAAT-3′) were used [67]. In our study, we used ITS5-1737F (5′-GGAAGTAAAAGTCGTAACAAGG-3′) and ITS2-2043R (5′-GCTGCGTTCTTCATCGATGC-3′) as primers [68]. The specific conditions and amplification procedure for the polymerase chain reaction (PCR) were as follows: first, a 1 min pre-denaturation at 98 °C; then 30 cycles, each consisting of denaturation at 98 °C for 10 s, annealing at 50 °C for 30 s, and extension at 72 °C for 30 s; and, finally, a 5 min extension at 72 °C. The PCR products were detected by 2% agarose gel electrophoresis. PCR products that passed the test were purified using magnetic beads and quantified using enzyme-linked immunosorbent assay. After quantification, the products were mixed proportionally according to the concentration. After thorough mixing, the PCR products were again assayed using 2% agarose gel electrophoresis. Subsequently, the target bands were recovered using the Gel Recovery Kit provided by QIAGEN. Libraries were constructed using the TruSeq^®^ DNA PCR-Free Sample Preparation Kit (Illumina, Inc., San Diego, CA, USA). The constructed libraries were quantified by Qubit and quantitative PCR. After the library was qualified, sequencing was performed using NovaSeq6000 [69].

### 4.5. Data Processing

The Uparse algorithm (Uparse v7.0.1001, http://www.drive5.com/uparse/), accessed on 28 July 2022 [70] was used to cluster all effective tags of all samples, and by default, the sequences were clustered into OTUs with 97% consistency; the Alpha diversity index was calculated using the Qiime (Version 1.9.1), and the differences in soil physicochemical properties, composition and diversity of soil microbial communities under different land uses were analyzed using SPSS software 25.0.with one-way analysis of variance (ANOVA) and LSD (*p* < 0.05) multiple comparisons of differences in soil physical and chemical properties, soil microbial community composition and diversity under different land use practices using SPSS software [71]. Spearman’s correlation coefficient method was used to analyze the correlation between soil properties and microbial community composition. The relative abundance of each soil microbial community was calculated according to different classification levels, and the microbial community composition was analyzed.

## 5. Conclusions

The present study revealed that in cedar plantation forests, bacterial and fungal communities were significantly clustered at a stand density of 900 trees ha^−1^, forming a clear distinction from communities at other densities. In addition, lower stand densities promoted soil nutrient regulation and increased bacterial diversity, contributing to the maintenance of a more complex and stable soil ecosystem structure, which is consistent with our initial hypothesis. These results provide a valuable reference for forest managers when developing silvicultural programs. Meanwhile, the results of the study can provide a theoretical basis for further understanding the distribution of soil microbial communities and their driving mechanisms under different stand densities.

## Figures and Tables

**Figure 1 plants-14-00098-f001:**
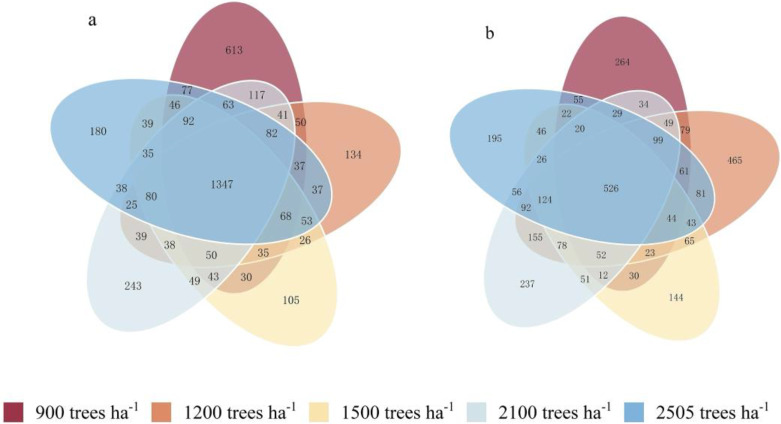
Venn diagrams of soil bacteria (**a**) and fungi (**b**) in Chinese fir plantations with different stand densities.

**Figure 2 plants-14-00098-f002:**
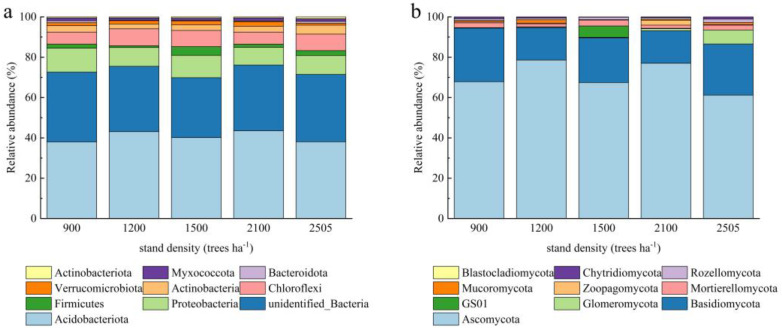
Effect of different stand densities of Chinese fir on relative abundance at the level of bacterial (**a**) and fungal (**b**) phyla.

**Figure 3 plants-14-00098-f003:**
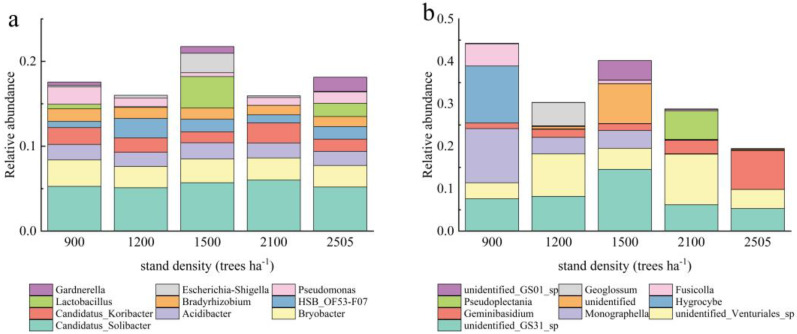
The effect of different stand densities of Chinese fir on the relative abundance of bacteria (**a**) and fungi (**b**) at the genus level.

**Figure 4 plants-14-00098-f004:**
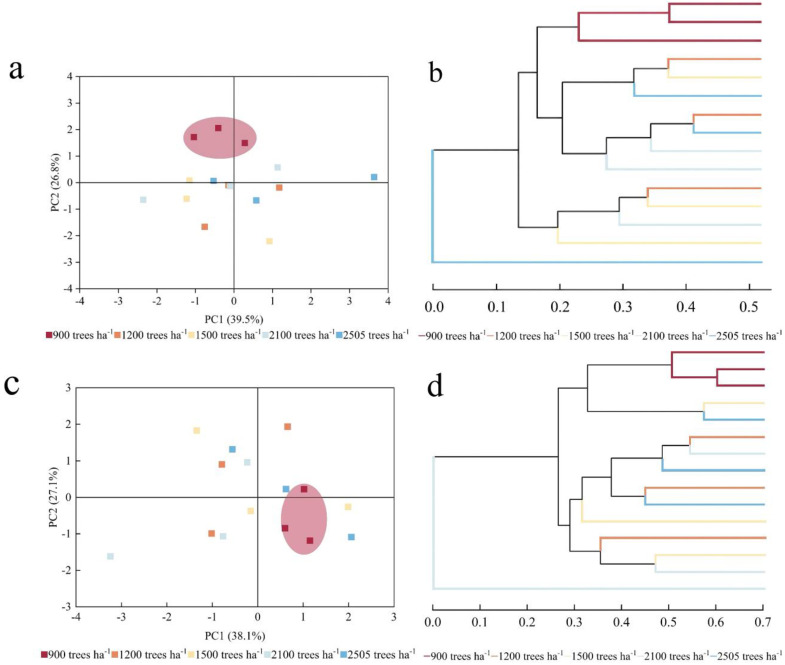
PCA analysis and hierarchical cluster analysis of soil microorganisms. The bacterial PCA analysis (**a**); the bacterial hierarchical cluster analysis (**b**); the fungal PCA analysis (**c**); the fungal hierarchical cluster analysis (**d**). Significant clustering was observed at a stand density of 900 trees ha^−1^, with ellipses highlighted.

**Figure 5 plants-14-00098-f005:**
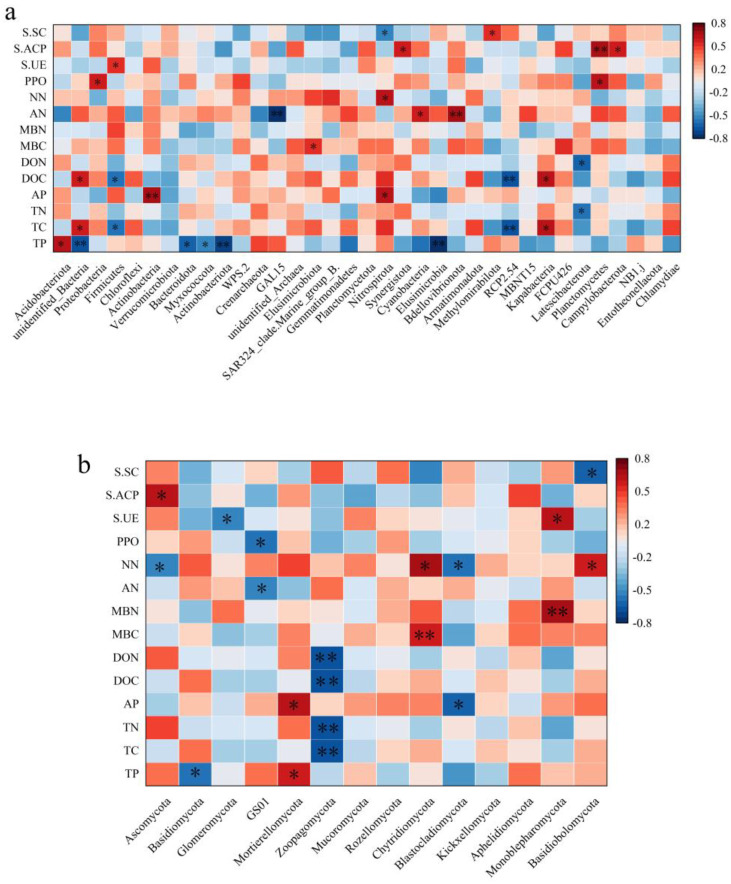
Correlation analysis between soil microbial communities and environmental factors in different stand densities. The bacterial (**a**) and fungal (**b**) communities. AN: ammonia nitrogen; NN: nitrate nitrogen; MBC: microbial biomass carbon; MBN: microbial biomass nitrogen; DOC: dissolved organic carbon; DON: dissolved organic nitrogen; PPO: polyphenol oxidase; S.UE: soil urease; S.ACP: soluble acid phosphatase; S.SC: soil sucrase (the same below). * denotes significant, ** denotes highly significant.

**Figure 6 plants-14-00098-f006:**
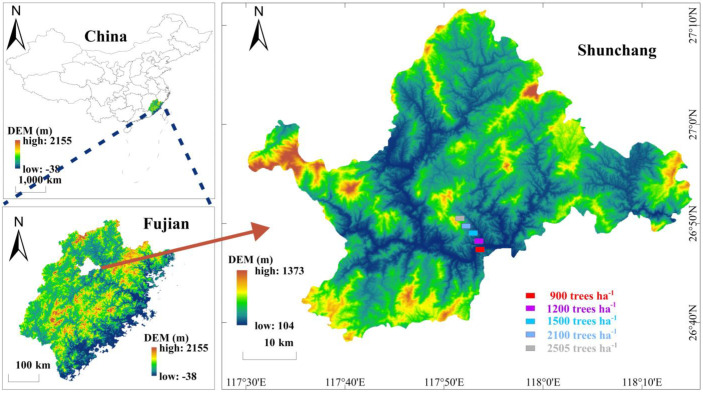
Overview of the research area.

**Table 1 plants-14-00098-t001:** Physical and chemical properties of soil in fir plantation forests with different stand densities.

Stand Density	900 Trees ha^−1^	1200 Trees ha^−1^	1500 Trees ha^−1^	2100 Trees ha^−1^	2505 Trees ha^−1^
TC (g/kg)	22.22 ± 1.79 a	17.99 ± 2.76 b	15.76 ± 1.41 c	20.14 ± 0.14 a	14.62 ± 2.08 d
TK (g/kg)	11.33 ± 2.5 b	15.2 ± 3.56 b	16.8 ± 3.13 ab	23.47 ± 4.43 a	10.8 ± 2.78 b
TN (g/kg)	2.12 ± 0.48 a	1.49 ± 0.22 a	1.98 ± 0.43 a	2.12 ± 0.48 a	1.95 ± 0.41 a
TP (g/kg)	0.52 ± 0.02 b	0.53 ± 0.01 b	0.59 ± 0.03 a	0.49 ± 0.03 b	0.50 ± 0.02 b
SOC (g/kg)	18.22 ± 0.01 b	19.12 ± 0.02 b	15.07 ± 0.03 d	18.2 ± 0.03 b	22.49 ± 0.06 a
SC (mg/g)	55.75 ± 6.98 c	71.04 ± 6.11 bc	82.59 ± 6.37 b	79.59 ± 4.49 b	98.48 ± 4.77 a
SN (mg/g)	37.24 ± 5.12 a	12.2 ± 1.1 b	18.68 ± 1.17 b	18.77 ± 1.36 b	16.46 ± 2.77 b
AK (mg/g)	342.23 ± 16.03 a	336.06 ± 6.59 a	335.5 ± 7.98 a	339.55 ± 11.38 a	342.18 ± 7.42 a
AP (mg/g)	3.22 ± 0.01 b	3.12 ± 0.02 c	3.07 ± 0.03 c	3.2 ± 0.03 b	3.49 ± 0.06 a
pH	4.73 ± 0.12 a	4.56 ± 0.14 a	4.53 ± 0.16 a	4.46 ± 0.3 a	4.36 ± 0.14 a

Note: Values are mean ± standard error. Different letters indicate significant differences between stand densities (*p* < 0.05) between stand densities. TC: total carbon; TK: total potassium; TN: total nitrogen; TP: total phosphorus; SOC: soil organic carbon; SC: soluble carbon; SN: soluble nitrogen; AK: available potassium; AP: available phosphorus.

**Table 2 plants-14-00098-t002:** Soil bacterial (a) and fungal (b) community alpha diversity indices in Chinese fir plantations with different stand densities.

	Stand Density	Shannon	Simpson	Chao	ACE	Good
Bacteria	900 trees ha^−1^	8.02 ± 0.09 a	0.98 ± 0.00 a	1957.03 ± 185.62 a	2018.97 ± 223.07 a	0.99 ± 0.00 a
	1200 trees ha^−1^	7.69 ± 0.21 a	0.98 ± 0.00 a	1600.61 ± 52.35 b	1624.40 ± 64.54 b	0.99 ± 0.00 a
	1500 trees ha^−1^	7.64 ± 0.39 a	0.98 ± 0.00 a	1597.40 ± 109.75 b	1622.14 ± 107.14 b	0.99 ± 0.00 a
	2100 trees ha^−1^	7.85 ± 0.02 a	0.98 ± 0.00 a	1708.64 ± 113.00 ab	1750.08 ± 119.79 ab	0.99 ± 0.00 a
	2505 trees ha^−1^	7.86 ± 0.10 a	0.98 ± 0.00 a	1733.00 ± 96.62 ab	1764.47 ± 106.82 ab	0.99 ± 0.00 a
Fungi	900 trees ha^−1^	4.97 ± 0.27 b	0.88 ± 0.04 b	825.08 ± 102.37 b	840.46 ± 106.60 b	0.99 ± 0.00 a
	1200 trees ha^−1^	6.59 ± 0.44 a	0.96 ± 0.01 a	1233.35 ± 104.88 a	1274.97 ± 94.16 a	0.99 ± 0.00 a
	1500 trees ha^−1^	5.20 ± 0.50 b	0.90 ± 0.04 b	750.34 ± 151.43 b	778.18 ± 161.26 b	0.99 ± 0.00 a
	2100 trees ha^−1^	6.00 ± 0.33 a	0.94 ± 0.00 a	982.24 ± 137.10 ab	1028.32 ± 156.18 ab	0.99 ± 0.00 a
	2505 trees ha^−1^	5.63 ± 0.14 b	0.98 ± 0.00 a	956.19 ± 175.48 ab	948.87 ± 152.93 ab	0.99 ± 0.00 a

**Table 3 plants-14-00098-t003:** Basic information of the survey sample site.

Stand Density	TH (m)	DBH (cm)	UBH (m)	CB (m)
900 trees ha^−1^	17.26 ± 0.79 a	21.68 ± 1.70 a	6.30 ± 0.80 b	6.97 ± 0.25 a
1200 trees ha^−1^	17.63 ± 0.89 a	19.71 ± 0.73 a	6.51 ± 0.35 b	6.97 ± 0.38 a
1500 trees ha^−1^	16.84 ± 1.40 a	21.22 ± 2.35 a	7.07 ± 1.43 ab	6.96 ± 0.10 a
2100 trees ha^−1^	16.43 ± 0.65 a	20.32 ± 1.93 a	8.50 ± 0.68 a	6.77 ± 0.41 a
2505 trees ha^−1^	16.60 ± 1.47 a	19.78 ± 2.08 a	8.21 ± 0.57 a	6.68 ± 0.76 a

Note: Different lowercase letters indicate significant differences in forest density between stands (*p* < 0.05). TH: tree height; DBH: diameter at breast height; UBH: under branch height; CB: crown breadth.

## Data Availability

Data are contained within the article.

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
