# Peer review of "Effects of Different Stand Densities on the Composition and Diversity of Soil Microbiota in a *Cunninghamia lanceolata* Plantation"

_plants, 2025, doi:10.3390/plants14010098_

Round 1
Reviewer 1 Report
Comments and Suggestions for Authors
The reviewed work has many cognitive values and the results can be used in practice. However, some corrections should be made.
The introduction discusses the importance of tree density on the activity of microorganisms and points out the scarcity of knowledge in this area. Therefore, the research aims should emphasize this. It would be good to list the aims and, best of all, put forward hypotheses. This will certainly be facilitated by conclusions that actually provide answers to possible hypotheses.
Line 113 – error, table 1 is discussed and not table 2, moreover there is no information about soil organic carbon (SOC) in the table
Line 123 – the table title is rather sparse. Please expand it. The table says- TC (g/kg) 22.22±1.79a 17..99±2.76b 15.76±1.41c 20.14±0.14a 14.62±2.08d , it should be TC (g/kg) 22.22±1.79a 17. 99±2.76b 15.76±1.41c 20.14±0.14a 14.62±2.08d
Line 131, section, 2.2.1 OTU cluster analysis – no reference to Figure 1 in the text. The same for line 153, section 2.2.2 Differences in the phylum-level structure of soil microorganisms, and Figure 2 ; line 176, section 2.2.3 Differences in genus-level structure of soil microorganisms and Figure 3; line 226, section 2.3.2 Principal component analysis and hierarchical cluster analysis and Figure 4; line 244, section 2.4. Correlation analysis between soil physical and chemical properties and microbial community and Figure 5.
Lines 422-423 – Please explain why the seedling density given in the methods was used in the experiment.
Line 427 – in the table, it is -1500 trees ha-1 16.84±1.40a 21.22±2.35a 7..07±1.43ab 6.96±0.10a, it should be -1500 trees ha-1 16.84±1.40a 21.22±2.35a 7.07±1.43ab 6.96±0.10a
Author Response
- The introduction discusses the importance of tree density on the activity of microorganisms and points out the scarcity of knowledge in this area. Therefore, the research aims should emphasize this. It would be good to list the aims and, best of all, put forward hypotheses. This will certainly be facilitated by conclusions that actually provide answers to possible hypotheses.
Response 1: thank you for your valuable advice. Based on your feedback, we have clearly outlined the objectives of the study in the introduction section and have formulated the following hypotheses. Moderate stand density promotes soil microbial diversity, which in turn has long-term positive effects on ecosystem services such as soil nutrient retention, carbon storage and biodiversity conservation. We believe this will help to emphasize the importance of the study and provide a solid basis for subsequent conclusions.
- Line 113 – error, table 1 is discussed and not table 2, moreover there is no information about soil organic carbon (SOC) in the table.
Response 2: thank you for pointing out the error. We have amended the reference in line 113 to ensure that Table 1 is being discussed rather than Table 2. In addition, we have added information about soil organic carbon (SOC) to the table to more fully reflect the results of the study.
- Line 123 – the table title is rather sparse. Please expand it. The table says- TC (g/kg) 22.22±1.79a 17..99±2.76b 15.76±1.41c 20.14±0.14a 14.62±2.08d , it should be TC (g/kg) 22.22±1.79a 17. 99±2.76b 15.76±1.41c 20.14±0.14a 14.62±2.08d
Response 3: thank you for your meticulous review. We have expanded the table headings to provide more detailed information. Multiple checks were made and ensured that the relevant data for the TC in the form was correct.
- Line 131, section, 2.2.1 OTU cluster analysis – no reference to Figure 1 in the text. The same for line 153, section 2.2.2 Differences in the phylum-level structure of soil microorganisms, and Figure 2 ; line 176, section 2.2.3 Differences in genus-level structure of soil microorganisms and Figure 3; line 226, section 2.3.2 Principal component analysis and hierarchical cluster analysis and Figure 4; line 244, section 2.4. Correlation analysis between soil physical and chemical properties and microbial community and Figure 5.
Response 4: thank you for pointing out the omission in the text regarding the citation of the chart. We have corrected the sections mentioned in the text to ensure that each section correctly references the corresponding chart. Here are the specific amendments: on line 131, we have added a reference to Figure 1, which is now explicitly mentioned in the text in relation to the OTU cluster analysis. On line 153, we have corrected this to a reference to Figure 2 and ensured that the discussion is related to the differences in soil microbial phylum structure in Figure 2. On line 176, we have added a reference to Figure 3 to reflect the analysis associated with differences in the genus-level structure of soil microbes. On line 226, we have corrected this to a reference to Figure 4 and ensured that principal component analysis and hierarchical cluster analysis are discussed. On line 244, we have added a reference to Figure 5 to show the correlation analysis between soil physicochemical properties and microbial communities. We have ensured that all chart references match the text content and have optimized the chart titles to provide clearer information. Thank you again for your careful review and valuable comments.
- Lines 422-423 – Please explain why the seedling density given in the methods was used in the experiment.
Response 5: thank you for your interest in our research. Regarding your question about initial planting density, we chose to use an initial planting density of 3300-3600 trees ha-1 in our experiments based on considerations of practice for medium-diameter or large-diameter fir cultivation. We have added an explanation of the choice of initial planting density in the "Materials and Methods" section of the article, and this modification will help the reader better understand our experimental design and the logic behind it.
- Line 427 – in the table, it is -1500 trees ha-1 16.84±1.40a 21.22±2.35a 7..07±1.43ab 6.96±0.10a, it should be -1500 trees ha-1 16.84±1.40a 21.22±2.35a 7.07±1.43ab 6.96±0.10a
Response 6: we apologize for this oversight and promise to carefully proofread subsequent manuscripts to avoid such errors from happening again. We appreciate your careful review, which helps to improve the accuracy and professionalism of our research reports.

Reviewer 2 Report
Comments and Suggestions for Authors
The article is devoted to the topical issue of the influence of tree stand density on the growth of tree species, understory diversity and soil properties. This research topic is of great importance, since in the management of forest ecosystems, stand density is recognized as a decisive factor affecting microbial diversity and soil functioning in fir stands. The authors showed that a sufficiently high number of Chinese fir (650 trees per hectare) promotes understory growth, increases species diversity and biomass, and improves the organic matter content and the ability of the soil to retain fertility, which has a positive effect on the stability and sustainable development of forests. It is known that optimal stand density is important for improving the ecological environment and species diversity of forest plantations and promotes community succession from forest plantations to natural forests. Moderate stand density promotes the growth of key functional groups of microorganisms, such as cellulolytic and ammonifiers, which play an important role in the decomposition of soil organic matter and the transformation of nutrients. However, high tree density can increase microbial diversity and abundance by providing more organic matter and habitat, promoting nitrogen and carbon cycling. Excessive tree density can lead to competition for resources and can affect the structure and function of the soil microbial community. Despite the large number of studies on this topic, how the soil microbiome responds to and adapts to different environmental factors under different forest density conditions has rarely been reported. The authors solve a small but specific problem that is valuable for forest management and knowledge of microbial ecology.
The article will be of interest to a wide range of researchers of forest ecosystems, soils, ecology and soil microorganisms. The authors' conclusions are based on the arguments of the obtained results and resolve the main questions posed in the study. References are appropriate and adequate. Of course, the authors have done a tremendous job of analyzing the samples, processing and interpreting the data. I highly appreciate this article, but I have a number of suggestions and comments for improving the article:
1 - The idea of the study is not clear. What are the authors' null hypothesis/hypotheses? There is no confirmation or refutation of the initial assumption of the study. The authors should add this information.
2 - The authors should expand the description of the objects under study, add more characteristics, and insert photos of the analyzed soils. The soil type according to various soil classifications is not indicated. There is no information on the type of land use of the territory and a detailed description of the soil samples. All these details are important for solving the research problems. The object of the study may not be entirely clear to readers.
3 – The number of physical replicates for the model laboratory experiment and for each of the methods is not specified. It may seem to the reader that the study was conducted without replicates, so some results seem unreliable.
4 – The authors report that the main dominants in all variants were Acidobacteria, Proteobacteria and Chloroflexi bacteria, as well as Ascomycota and Basidiomycota fungi. However, in the abstract, the authors write: "Our results show that planting density is a key factor determining soil microbial diversity." Thus, the authors contradict themselves, since the planting density was different in different variants, but the taxonomic composition of microorganisms was the same. In addition, the obtained results on diversity are quite trivial - such taxonomic diversity of microorganisms is typical for most soils and is not specific.
5 – Table 1 has a reduced title that needs to be expanded.
6 – Section "4. Materials and Methods» is located at the end of the scientific article. However, it would be more rational to move this section to section «2. Results». This is necessary, since it is important for the reader to familiarize himself with the objects and methods before reading the results.
7 – The authors should expand section «5. Conclusions», which currently looks poor and does not fully disclose the solution to the research problem. It would be important for a wide range of readers to see specific recommendations on the planting density of Cunninghamia lanceolata trees in section «5. Conclusions». It would also be useful to indicate a similar justification for choosing a certain number of trees per hectare.
8 – Literary references often contain errors and are not always formatted according to the journal requirements.
Author Response
- The idea of the study is not clear. What are the authors' null hypothesis/hypotheses? There is no confirmation or refutation of the initial assumption of the study. The authors should add this information.
Response 1: thank you for your feedback and suggestions. It's vital for us to improve our articles.Based on your comments, we have clarified our research hypotheses in the introduction section and tested them in our study.We believe that by clearly formulating the research hypothesis and testing it, our article will not only be able to communicate the purpose of the study more clearly, but also be able to interpret and discuss the findings more effectively.
- The authors should expand the description of the objects under study, add more characteristics, and insert photos of the analyzed soils. The soil type according to various soil classifications is not indicated. There is no information on the type of land use of the territory and a detailed description of the soil samples. All these details are important for solving the research problems. The object of the study may not be entirely clear to readers.
Response 2: thank you for your valuable advice. We have revised the article accordingly to enhance the description of the objects under study. We added the specific location and topographic features of the study area, clarified the soil type as montane red loam, and provided detailed information on the soil's organic matter content, depth, pH, and stand index. In addition, we illustrated the area as a plantation timber forest that has been established for more than 20 years and classified it accordingly based on land use classification. This additional information helps the reader to understand the context and object of the study more clearly and ensures clarity of the research question.
- The number of physical replicates for the model laboratory experiment and for each of the methods is not specified. It may seem to the reader that the study was conducted without replicates, so some results seem unreliable.
Response 3: we understand that in scientific research, the number of replicate samples is critical to the reliability and validity of results. To ensure the accuracy and reproducibility of our results, we have added in 4.2 Sample design and sample collection that each sample was repeated three times for soil microbial community analysis. We believe that with these additions, our study has become more methodologically rigorous and our results more reliable.
- The authors report that the main dominants in all variants were Acidobacteria, Proteobacteria and Chloroflexi bacteria, as well as Ascomycota and Basidiomycota fungi. However, in the abstract, the authors write: "Our results show that planting density is a key factor determining soil microbial diversity." Thus, the authors contradict themselves, since the planting density was different in different variants, but the taxonomic composition of microorganisms was the same. In addition, the obtained results on diversity are quite trivial - such taxonomic diversity of microorganisms is typical for most soils and is not specific.
Response 4: thank you for your careful review and valuable comments. We understand your concern that the article's description of the effect of planting density on soil microbial diversity may contradict the consistency of microbial composition mentioned in the results section. To address the contradictory questions you have raised, we have further elaborated in our article on how stand density affects soil microbial diversity by influencing the relative abundance and diversity indices of the microbial community. Our results suggest that changes in soil physicochemical properties and microenvironmental conditions are key drivers of soil microbial diversity. Finally, we agree with you that the taxonomic diversity of soil microorganisms is typical in most soils and not specific. We hope that these modifications and additions will address your concerns and enhance the persuasiveness of our study. Thank you again for your valuable comments.
- Table 1 has a reduced title that needs to be expanded.
Response 5: we have expanded the table headings to provide more detailed information.
- Section "4. Materials and Methods» is located at the end of the scientific article. However, it would be more rational to move this section to section «2. Results». This is necessary, since it is important for the reader to familiarize himself with the objects and methods before reading the results.
Response 6: we understand your concern about the location of the Materials and Methods section. However, according to the formatting requirements of our submission, the Results section is located before the Materials and Methods section. We follow the formatting requirements of the journals to ensure that the articles meet the publication standards. We believe that even though the "Materials and Methods" section is located after the "Results", we have clearly labeled the sections in the text and made it possible for the reader to make a smooth transition from the results to the understanding of the methodology through a logical and clear narrative. We have also ensured that the key findings and data mentioned in the Results section can be found in the subsequent Materials and Methods section, where details of the experimental design and manipulation can be found for easy comparison and understanding by the reader.
- The authors should expand section «5. Conclusions», which currently looks poor and does not fully disclose the solution to the research problem. It would be important for a wide range of readers to see specific recommendations on the planting density of Cunninghamia lanceolata trees in section «5. Conclusions». It would also be useful to indicate a similar justification for choosing a certain number of trees per hectare.
Response 7: thanks to your valuable comments, we recognize the importance of the conclusion section and its key role in presenting the findings and solutions in a comprehensive manner. In response to your suggestions, we have expanded and deepened the «5. Conclusions» to ensure fuller disclosure of the solutions to the research questions and to provide the reader with concrete recommendations.
- Literary references often contain errors and are not always formatted according to the journal requirements.
Response 8: based on your feedback, we have thoroughly reviewed and corrected the literature citations.
